# Physical Self-Concept and Physical Activity in Children with Congenital Heart Defects—Can We Point Out Differences to Healthy Children to Promote Physical Activity?

**DOI:** 10.3390/children10030478

**Published:** 2023-02-28

**Authors:** Jannos Siaplaouras, Annika Jahn, Paul Helm, Kerstin Hanssen, Ulrike Bauer, Christian Apitz, Claudia Niessner

**Affiliations:** 1Praxis am Herz-Jesu-Krankenhaus, 36037 Fulda, Germany; 2Division of Pediatric Cardiology, Children’s Hospital, University of Ulm, 89073 Ulm, Germany; 3National Register for Congenital Heart Defects, DZHK (German Centre for Cardiovascular Research), 13353 Berlin, Germany; 4Institute for Sport and Sport Science, Karlsruhe Institute for Technology, 76131 Karlsruhe, Germany

**Keywords:** self-concept, movement behavior, heart defect, youth

## Abstract

Objective: Children with congenital heart defects (CHD) are at high risk for cardiovascular disease in addition to their congenital disease, so it is important to motivate this group of patients to live a physically active lifestyle. A potential influencing determinant of younger children’s physical performance is the physical self-concept. The objective of the present study was first to evaluate the correlation between the physical self-concept (PSC) and the participation in physical activities (PA) of a representative group of children with congenital heart disease (CHD), and second to point out differences in comparison to their healthy peer group. Methods: Using the database of PA of the S-BAHn-Study we focused on physical self-concept assessed by the German version of the Physical Self-Description Questionnaire. We compare the obtained data of children with CHD to a representative age-matched sample of 3.385 participants of the Motorik Modul Study. Results: N = 1.198 complete datasets could be included in the analyses. The mean age of patients was 11.6 ± 3.1 years. For the total cohort of patients with CHD and the reference group, PA correlated significantly with a positive PSC (*p* < 0.001). PA was significantly reduced in all groups of patients despite the severity of their heart defect (*p* < 0.001). Remarkably, PSC did not differ statistically significantly in patients with simple CHD from the reference collective (*p* > 0.24). Conclusions: According to this representative survey, there is a clear relation between PA and PSC in the cohort of healthy children and the group of children with CHD throughout the severity of their heart defects. Although PSC did not differ in patients with simple CHD and their healthy peer group, PA was significantly reduced. This gap invites us to reflect on how we could break new ground to promote a physically active lifestyle in children with CHD regardless of the severity of their cardiac defects.

## 1. Introduction

Congenital heart defects are the most frequent congenital deformation in childhood. They occur in 1.1% of in newborns [1]. Today, technical progress has reached the point where medical care and surgical methods exist that allow children with CHD to grow up very well [2]. However, in affected children, as in healthy children and adolescents, cardiovascular risk factors (i.e., cardiovascular disease) increase [3,4]. Myocardial infarction could become the major cause of death in CHD patients with simple cardiac defects [5].

This highlights the need for primary prevention. Physical activity (PA) is the key to well-being and a healthy life, especially in early childhood when children’s physical abilities mainly develop [6].

In general, only 12% of children and adolescents achieve the WHO-recommended physical activity of at least 60 min per day [7]. In comparison, only 8.8% of children and adolescents with CHD achieve these recommendations [8]. The lack of physical activity has several causes. Bjarnason-Wehrens [9] posited the “vicious cycle” as an explanatory approach. The lack of physical activity is created by overprotection and anxiety in the family of the affected person. This is reinforced by motor deficits. Motor deficits, in turn, increase anxiety and overprotection in the family. A “vicious cycle” develops. In addition, psycho-social situations become more difficult and the family’s radius of action is limited. 

To address the problem of overprotection, there are recommendations for assessing the physical fitness of children with CHD and detailed recommendations for physical activity even for competitive athletes with cardiovascular abnormalities [10,11,12,13]. Regarding actual recommendations, most patients with CHD are allowed to participate in competitive and leisure sports, unrestricted like healthy children. Only patients who are about to undergo surgery or who have life-threatening findings are not allowed to do sports [10]. Otherwise, all patients are allowed to do sports. However, in patients with severe or significant (residual) findings, attention should be paid to the type and load of the sport.

One factor that influences children’s physical performance (e.g., sports participation) is the physical self-concept (PSC) [14]. The knowledge of this motivational process could thus be of great importance for improving PA. Therefore, research is of great interest in children with CHD.

The body of studies on physical self-concept in children with a congenital heart defect is very limited [15,16,17]. Only the studies by Chen et al. [16] used a questionnaire specific to the physical self-concept, the Physical Self-Description Questionnaire (PSDQ). In the other two studies [15,17], physical self-concept was asked as a subscale in a questionnaire used for general self-concept.

Chen et al. [15,17] only made comparisons among children with CHD. Thereby, both studies found a significantly better physical self-concept among boys compared to girls.

Chen et al. [16] analyzed the difference between school children with CHD and healthy school children. They found significantly lower physical self-concept scores in children with CHD compared to healthy children. However, due to its small sample size, the study by Chen et al. [16] is only of limited significance for the overall population.

Using the unique database of the S-Bahn Study [8], we aimed (I) to obtain representative data regarding the physical self-concept in children with CHD and its correlation to the severity of the heart defect; (II) to correlate PSC to the amount of PA; and (III) to detect differences between a representative group of children with and without CHD.

## 2. Materials and Methods

### 2.1. Study Design

Ethical approval was obtained from Charité, Berlin (Approval number 2/034/17) and the Karlsruhe Institute of Technology. The study protocol complies with the ethical guidelines of the 1975 Declaration of Helsinki. This study is a cross-sectional study based on the collection of data from January to March 2018 through questionnaires. The data are composed of two studies. One study included subjects with CHD (S-BAHN study) and the other study is available as a reference group (MoMo study). Within the S-BAHN study, we contacted patients via the patient database of the NRCHD, the largest registry for CHD patients in Europe.

### 2.2. Survey Instruments

The validated questionnaire of the “Motorik-Modul” Study (MoMo), which is an in-depth study of the German Health Interview and Examination Survey for Children and Adolescents (KiGGS), was used to assess PSC and PA. The MoMo study has been recording the PA of children and adolescents at regular intervals of 5 years since 2003, and thus provides an important basis for a health monitoring of children in Germany. Using the MoMo, PAQ allowed us to compare the collected data with a representative age-matched reference collective of 3385 participants from the MoMo Wave 2 study (2015–2017) [18]. 

Further information, i.e., results and design of the MoMo baseline and longitudinal studies, is published elsewhere [19,20,21,22]. Detailed information about the MoMo Physical Activity Questionnaire (MoMo-PAQ), such as validity, content, and details on the structure, was published in high-ranking journals [19,20,21,22]. 

#### 2.2.1. Physical Activity (PA) 

The MoMo PAQ contains questions with single or multiple answers, and consists of 28 items on frequency, duration, and intensity of physical activity to assess habitual physical activity in different domains, for example physical activity in sports clubs, leisure time activity outside sports clubs, extracurricular physical activity, playing outdoors, and active commuting to school. 

#### 2.2.2. Physical Self-Concept (PSC)

PSC was analyzed by using 36 items of the MoMo-PAQ, which assesses physical self-description based on the German version of the Physical Self-Description Questionnaire with response categories on a 4-point Likert scale [22]. These 36 items represent the basic functions of physical performance, namely, overall sporting skills (here referred to as “skill”), strength, endurance, speed, flexibility, and coordination.

### 2.3. Statistical Analysis

Values of continuous variables are reported as mean ± standard deviation. Pearson’s chi-square test was used for group comparisons including nominal data (e.g., gender and age). In order to estimate the impact of potential contributing factors on PA, analysis of variance and covariance, Pearson’s correlation, and multiple and linear regression analysis were used. IBM SPSS statistics version 25.0 (IBM Inc., Armonk, NY, USA) was used for statistical analyses. A significance level of *p* ≤ 0.05 was applied.

## 3. Results

### 3.1. Patient Characteristics

Of 21.354 eligible patients, the invitation was successfully delivered to 14.496 patients of whom 1.718 patients participated in the study. 

N = 1.198 complete datasets could be included in the analyses. The mean age of patients was 11.6 ± 3.1 years, we could include 46.2% females and 53.8% males (Table 1 and Table 2). 

The patient characteristics of the study group were consistent with the overall cohort of registered eligible patients (Table 3). In the patient group, 47% report that their parents had a high-school graduation (fathers (47%) and mothers (47.4%)), and in the reference collective 45.8% of mothers and 43.9% of fathers had a high-school graduation (Table 4). In total, 57.2% of patients reported living in an urban or suburban area, whereas 42.8% lived in a rural environment. 

We used Warnes’ categorization [23] to divide study participants into simple, moderate, and complex CHD. In total, 34.3% (N = 411) were classified as simple CHD, 35.3% (N = 423 moderate, and 30.4% (N = 364) as complex CHD. Additionally, 20.5% of the patients had more than three operations or other interventions, 30.2% had 1–3 operations or other interventions, and the majority (49.3%) had untreated CHD.

Genetic syndromes were present in 5.8% (N = 70), most frequently trisomy 21 with 3.1% (N = 37), and Di-George syndrome in 1.3% (N = 15) (Table 3). 

Table 1 modified according to Jahn, Annika (2022): Körperliche Aktivität und Sportverhalten bei Kindern und Jugendlichen mit angeborenen Herzfehlern—eine deutschlandweite Analyse. Universität Ulm. Dissertation. http://dx.doi.org/10.18725/OPARU-46553 (accessed on 13 October 2022).

**Table 2 children-10-00478-t002:** CHD classification according to Warnes et al. [24].

	Simple CHD	Moderate CHD	Complex CHD
Participants (n)	411	423	364
Participants (in percent)	34.3	35.3	30.4
Mean age *	11.18 ± 2.92 Min.: 6; Max.: 17	11.84 ± 3.04 Min.: 6; Max.: 17	11.64 ± 3.19 Min.: 6; Max.: 17
Gender (m:f) in %	46:54	55.1:44.9	61:39
Operations (n) (Min.: 0; Max.: 18)	0.19 ± 0.63 Min.: 0; Max.: 5	1.4 ± 1.65 Min.: 0; Max.: 10	4.26 ± 3.26 Min.: 0; Max.: 17

* mean ± standard deviation.

Table modified according to Jahn, Annika (2022): Körperliche Aktivität und Sportverhalten bei Kindern und Jugendlichen mit angeborenen Herzfehlern—eine deutschlandweite Analyse. Universität Ulm. Dissertation. http://dx.doi.org/10.18725/OPARU-46553 (accessed on 13 October 2022)

**Table 3 children-10-00478-t003:** Frequency distribution of genetic syndromes in the patients cohort.

Genetic Syndrome	%
Down Syndrome	3.1
Di-George Syndrome	1.3
CHARGE Syndrome	0.3
Chromosomal anomaly	0.3
Noonan Syndrome	0.2
VACTERL Association	0.2
Goldenhar Syndrome	0.1
Turner Syndrome	0.1
Williams-Beuren Syndrome	0.1
Klinefelter Syndrome	0.1

Table modified according to Jahn, Annika (2022): Körperliche Aktivität und Sportverhalten bei Kindern und Jugendlichen mit angeborenen Herzfehlern—eine deutschlandweite Analyse. Universität Ulm. Dissertation. http://dx.doi.org/10.18725/OPARU-46553 (accessed on 13 October 2022)

**Table 4 children-10-00478-t004:** Educational qualifications of parents (patients vs. reference).

	Patients (%)	Reference (%)
Father without graduation	0.5	0.7
Father completed secondary school	16.5	16.2
Father completed comprehensive school	31.3	35.5
Father with high-school graduation	47.0	45.8
Father with other educational qualifications	4.7	1.8
Mother without graduation	0.8	0.4
Mother completed secondary school	8.4	9.1
Mother completed comprehensive school	40.5	43.9
Mother with high-school graduation	47.4	44.9
Mother with other educational qualifications	2.9	1.7

Table modified according to Jahn, Annika (2022): Körperliche Aktivität und Sportverhalten bei Kindern und Jugendlichen mit angeborenen Herzfehlern—eine deutschlandweite Analyse. Universität Ulm. Dissertation. Open Access Repositorium der Universität Ulm und Technischen Hochschule Ulm. Dissertation. http://dx.doi.org/10.18725/OPARU-46553 (accessed on 13 October 2022).

### 3.2. Physical Activity

According to WHO recommendations, children and adolescents should carry out at least an average of 60 min per day of moderate-to-vigorous intensity, mostly aerobic physical activity throughout the week [24].

Compared to the reference group, the total cohort of CHD patients was outstandingly less physically active (3.4 vs. 4 days physical activity per week, *p* < 0.001) and reached the level of 60 min of daily PA demanded by the WHO significantly less often (8.8 vs. 12%; *p* < 0.001). Total weekly activity decreased as the severity of the heart defect increased. Children with simple CHD were physically active 3.5 days per week, with moderate CHD 3.4 days, and with complex CHD 3.3 days (Figure 1 and Figure 2).

**Figure 1 children-10-00478-f001:**
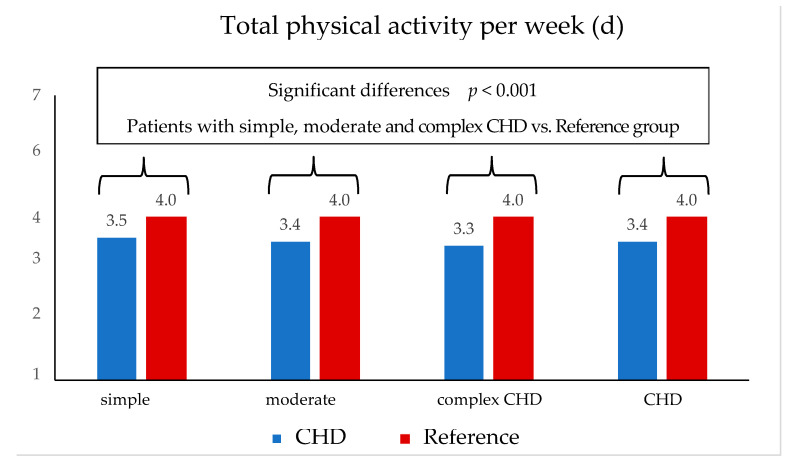
Total physical activity in days per week (patients vs. reference).

Figure 1 modified according to Jahn, Annika (2022): Körperliche Aktivität und Sportverhalten bei Kindern und Jugendlichen mit angeborenen Herzfehlern—eine deutschlandweite Analyse. Universität Ulm. Dissertation. Open Access Repositorium der Universität Ulm und Technischen Hochschule Ulm. Dissertation. http://dx.doi.org/10.18725/OPARU-46553 (accessed on 13 October 2022).

**Figure 2 children-10-00478-f002:**
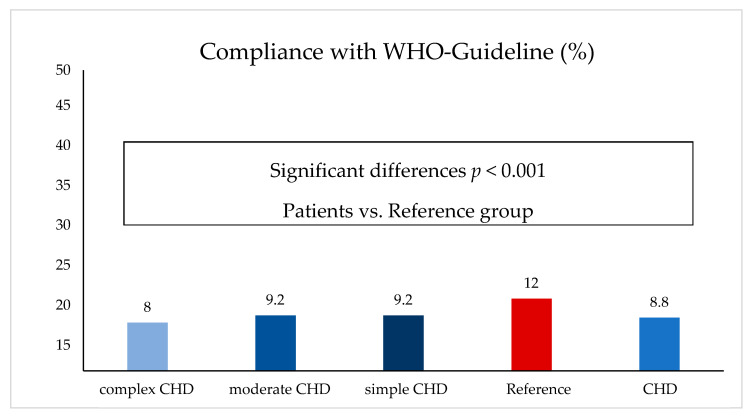
Compliance with WHO guidelines (patients vs. reference).

Figure 2 modified according to Jahn, Annika (2022): Körperliche Aktivität und Sportverhalten bei Kindern und Jugendlichen mit angeborenen Herzfehlern—eine deutschlandweite Analyse. Universität Ulm. Dissertation. Open Access Repositorium der Universität Ulm und Technischen Hochschule Ulm. Dissertation. http://dx.doi.org/10.18725/OPARU-46553 (accessed on 13 October 2022).

### 3.3. Physical Self-Concept

The total patient cohort rated their ability to perform best in the categories of coordination and flexibility (56.6% and 56% respectively). The worst result was achieved in the category of endurance. Only 28.5% of all patients with CHD stated a positive self-concept for this item (Figure 3).

The proportion of the patients with simple CHD reached a positive self-concept in all categories and did not differ significantly in any of the categories compared to the reference group (post hoc test: *p* > 0.24). Remarkably, patients with simple CHD rated their athletic performance even better than the reference group in the categories of coordination (68% vs. 64.9%) and mobility (62.1% vs. 56.8%) (Table 5).

The participants with moderate CHD showed notably different mean values in each category (post hoc test: *p* < 0.05), except for the categories of strength (post hoc test: *p* > 0.17) and mobility (post hoc test: *p* > 0.57). 

Patients with complex CHD consistently rated themselves as worst and achieved significantly lower mean values in all categories than all other subgroups (post hoc test: *p* < 0.01). They achieved a positive self-concept least often in the endurance category (16.8%).

### 3.4. Physical Self-Concept and Physical Activity

PA correlated significantly with the achievement of a positive PSC. Respondents with simple CHD achieved the strongest correlation coefficients between total activity and physical self-concept. The higher the total physical activity was, the higher the achieved values in the physical self-concept and the more often a positive PSC was achieved. This correlation was clearest for the dimension “endurance” (Figure 3 and Table 5). The data showed that only 28.5% of patients with CHD achieved a positive PSC in the category “endurance”. This result makes it clear that low endurance is a painful limitation for patients with CHD. Not least for this reason, dynamic sports that train endurance skills are particularly recommended for children and adolescents with CHD.

On further analysis of the subgroups, it was noticeable that there was a difference between the severity of heart defect in terms of PSC. While a large proportion of patients with simple CHD achieved a positive PSC in all categories and the patients with simple CHD did not differ significantly from the reference collective in any category, the patients with complex CHD differed significantly from the other subgroups in the frequency of positive PSC in all categories and achieved a positive PSC less frequently than the other subgroups in all categories.

Figure 3 modified according to Jahn, Annika (2022): Körperliche Aktivität und Sportverhalten bei Kindern und Jugendlichen mit angeborenen Herzfehlern—eine deutschlandweite Analyse. Universität Ulm. Dissertation. Open Access Repositorium der Universität Ulm und Technischen Hochschule Ulm. Dissertation. http://dx.doi.org/10.18725/OPARU-46553 (accessed on 13 October 2022).

**Figure 3 children-10-00478-f003:**
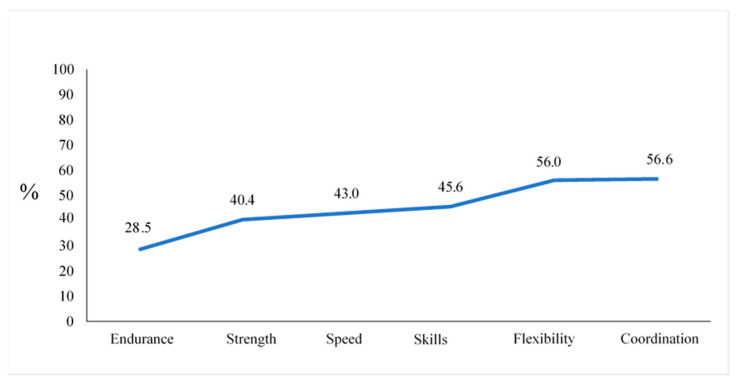
Frequency of a positive self-concept in the CHD group (in percent).

**Table 5 children-10-00478-t005:** Frequency of a positive self-concept per CHD group and reference.

		Complex CHD	Moderate CHD	Simple CHD	Reference
Endurance	Positive PSC (%)	16.8	29.2	39.9	45.1
	n=	214	267	203	2103
	*p*	<0.001	<0.001	0.55	
Coordination	Positive PSC (%)	43.9	58.1	68.0	64.9
	n=	214	267	203	2070
	*p*	<0.001	0.05	0.35	
Skills	Positive PSC (%)	30.8	47.2	59.1	62.2
	n=	214	267	203	2086
	*p*	<0.001	<0.001	0.68	
Strength	Positive PSC (%)	29.4	43.4	47.8	47.8
	n=	214	267	203	2075
	*p*	<0.001	0.46	0.55	
Flexibility	Positive PSC (%)	43.5	61.4	62.1	56.8
	n=	214	267	203	2100
	*p*	<0.001	0.99	0.24	
Speed	Positive PSC (%)	30.4	44.2	54.7	57.3
	n=	214	267	203	2090
	*p*	<0.001	<0.001	0.99	

Statistics: CHD group (complex, moderate, simple) vs. reference.

Table modified according to Jahn, Annika (2022): Körperliche Aktivität und Sportverhalten bei Kindern und Jugendlichen mit angeborenen Herzfehlern—eine deutschlandweite Analyse. Universität Ulm. Dissertation. Open Access Repositorium der Universität Ulm und Technischen Hochschule Ulm. Dissertation. http://dx.doi.org/10.18725/OPARU-46553 (accessed on 13 October 2022).

Figure 4 modified according to Jahn, Annika (2022): Körperliche Aktivität und Sportverhalten bei Kindern und Jugendlichen mit angeborenen Herzfehlern—eine deutschlandweite Analyse. Universität Ulm. Dissertation. Open Access Repositorium der Universität Ulm und Technischen Hochschule Ulm. Dissertation. http://dx.doi.org/10.18725/OPARU-46553 (accessed on 13 October 2022).

**Figure 4 children-10-00478-f004:**
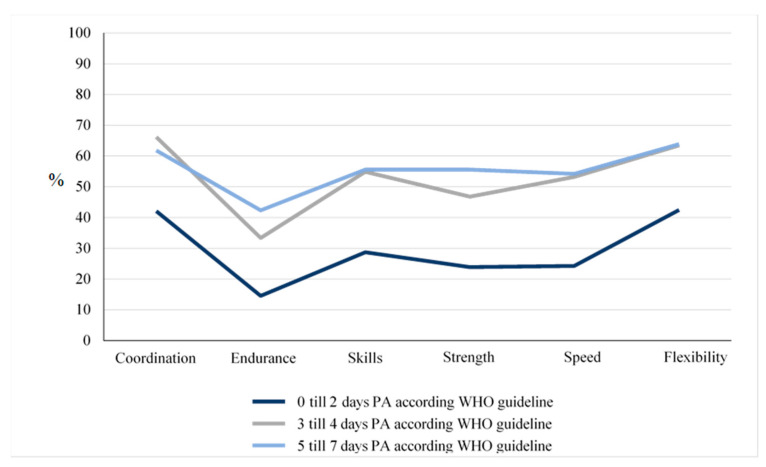
Relationship between a positive self-concept and physical activity in the CHD group.

There is a clear relationship between a positive physical self-description (%) of basic functions of physical performance and the level of PA reported in days per week (Figure 4 and Table 6). The percentage of patients reporting a positive self-description is significantly higher for those who are physically active on more than two days per week.

**Table 6 children-10-00478-t006:** Correlation of a positive self-concept with overall activity (per CHD group).

	Endurance	Coordination	Skills	Strength	Flexibility	Speed
Complex CHD	0.25 *	0.23 *	0.28 *	0.28 *	0.21 *	0.25 *
Moderate CHD	0.33 *	0.27 *	0.34 *	0.33 *	0.24 *	0.29 *
Simple CHD	0.41 *	0.36 *	0.45 *	0.38 *	0.36 *	0.33 *
Reference	0.34 *	0.28 *	0.34 *	0.31 *	0.20 *	0.27 *

* Correlation is significant at a level of 0.01 (both sides). For each CHD group and the reference group the correlation between each category of the self-concept and the physical activity was significant.

Table modified according to Jahn, Annika (2022): Körperliche Aktivität und Sportverhalten bei Kindern und Jugendlichen mit angeborenen Herzfehlern—eine deutschlandweite Analyse. Universität Ulm. Dissertation Open Access Repositorium der Universität Ulm und Technischen Hochschule Ulm. Dissertation. http://dx.doi.org/10.18725/OPARU-46553 (accessed on 13 October 2022).

## 4. Discussion

Who am I? This question plays an important role in adolescence. It is a crucial contributor to the developmental process not only in healthy children and adolescents but also in children and adolescents with CHD. However, due to the disease burden, these children usually face more difficult challenges in this process, such as dealing with physical limitations [23,25,26].

Adequate physical activity is important for socio-emotional and physical development and a healthy life and is crucial to prevent affluence diseases. However, to the best of our knowledge, no research has evaluated PSC and PA simultaneously in a representative cohort of healthy children and children with CHD. Our study strived to explore the relations between PSC and PA.

As expected, there was a correlation between the severity of the heart defect and the PSC of the participants. PSC correlated with the amount of PA. More precisely, this effect was statistically significant in healthy children, regardless of the severity of the heart defect for children with simple, moderate, and severe CHD. Viewed from another perspective, reduced PA correlated with an impaired PSC. 

Jekauc et al. [19] described physical self-concept as one of the determinants of physical activity in children. Thus, motivational strategies to increase positive physical self-concept and strategies to increase self-awareness may be important for maintaining physical activity, especially during the transition from adolescence to adulthood. Increasing physical self-concept can be realized with the help of psychological interventions or through specific thematization within a sports intervention program.

Children with complex CHD were less physically active than their peers with CHD. Unexpectedly, children with simple CHD, which means children who normally are allowed to do unrestrictive leisure and competitive sports, were also notably less physically active compared to the healthy reference group. Interestingly, children with simple CHD did not differ in any of the PSC categories significantly from the reference group and rated their physical performance in the categories of coordination and flexibility even better than the reference group. 

Complementary to these results, it is important to state that children with CHD enjoy physical activity on a comparable level as the reference group of healthy children as published before [8]. 

Psychosocial as well as physical causes could be the reason for the differing results of patients with simple and complex CHD. One psychosocial reason for the difference in physical self-concept could be that the respondents with simple CHD were more likely to participate in sports with healthy people of the same age or with their peer group. For patients with complex heart defects, sports with peers could be hampered by several factors, including increased physician-advised sports restrictions, the severity of the heart defect and the associated reduced cardiopulmonary performance, or the lack of infrastructure in the sports club that would allow the integration of patients with complex heart disease into sports clubs with peers.

Returning to the initial question, according to this study we are able to point out differences between healthy children and those with CHD. Preserved PSC cannot be considered an independent explanatory factor for the gap in PA in the cohort of children with simple CHD compared to their peer group. The bottom line is that, since physical activity is multifactorial, it is not surprising that self-concept is not sufficient to predict activity. We would certainly also need to know more about environmental factors (social support, clubs, playgrounds, parental attitudes, etc.).

The presented intention–behavior gap in CHD patients implies that we should additionally focus on general conditions for sporting activities, such as new concepts in school and club sports and individualized advice. However, the major implication arising from this study is to promote impaired PA irrespective of the severity of CHD and beyond the burden of the heart defect itself.

### Strength and Limitation

For this study, a questionnaire was chosen to collect the data. Questionnaires are suitable for quantitative interviewing because they are associated with low effort and costs. The disadvantage of questionnaires is that they are rather general and, in contrast to interviews, less detailed and cannot respond to the individual. In return, the influence of unintentionally and unconsciously dealing with each person differently is eliminated, resulting in deviations in measurement accuracy. Furthermore, the psychological effect of social desirability can lead to falsification of the answers [27]. The CHD patients were contacted via an online questionnaire and therefore had no opportunity to ask comprehension questions directly. To counteract this, participants had the option of calling the study leader or documenting comprehension questions at the end. For more accurate physical activity measurement data, directly measured physical activity using an accelerometer, for example, would be preferable [28]. However, this would lead to a significantly higher financial outlay and a smaller sample size would be the consequence. Questionnaires are therefore a suitable means of survey for large samples for this research question.

One aspect that was not surveyed in this work is BMI. Due to the high number of overweight and obesity in CHD patients (Pinto et al., 2007; Tamayo et al., 2015) and the known influence on body perception and physical self-concept in healthy children and adolescents (Fernández-Bustos et al., 2019), it is an important parameter in terms of physical self-concept but could not be considered in this work.

This work is a cross-sectional analysis. Thus, causal relationships cannot be derived. However, a cross-sectional analysis is sufficient for the presentation of descriptive information.

It cannot be ruled out that the reference group also includes sick children. However, since this concerns only a minority, there should be no bias from a statistical point of view.

Due to the large sample size and the cooperation with the NRCHD and the Motorik Modul study, representative data for Germany are presented.

## 5. Conclusions

Our results from this nationwide survey suggest that PSC and PA are deeply connected in children with CHD. An improvement in one leads to an increase in the other and vice versa. However, since PA is multifactorial, the self-concept is not sufficient enough to predict individual activity.

The circumstances leading to the outlined PA gap in children with simple CHD compared to their peer group are neither caused by impaired PSC nor by reduced enjoyment in sports. Possible factors causing this gap are framework conditions for sporting activities in Germany, such as a lack of new concepts in school and club sports and individualized advice for this group of children.

To avoid a sedentary lifestyle, we should focus on these matters in further studies. 

## 6. Patents

This section is not mandatory but may be added if there are patents resulting from the work reported in this manuscript.

## Figures and Tables

**Table 1 children-10-00478-t001:** Age and gender of patients and the reference group.

	Patients	Reference
Participants	1198	3385
Mean age *	11.55 ± 3.06; Min.: 6; Max.: 17	11.97 ± 3.39; Min.: 6; Max.: 17
Gender (m:f) in %	53.8:46.2	49.7:50.3

* mean ± standard deviation.

## Data Availability

Additional data and SPSS software code can be obtained from the authors.

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
