# Peer review of "Physical Self-Concept and Physical Activity in Children with Congenital Heart Defects—Can We Point Out Differences to Healthy Children to Promote Physical Activity?"

_children, 2023, doi:10.3390/children10030478_

Round 1

Reviewer 1 Report

1) I think the abstract is too long. Please mention most important issues on the abstract. For example, answer categories of the questionnaires are not really important in the abstract.

2) Introduction needs to be extended, mostly using reporting the previous studies regarding the levels of PA in children with CHD as well as the effects of some important psychological factors on participation of children with CHD in PA.

3) I think it is better to place the tables and figures in right place (i.e., where the data are talked about).

4) Please explain about the strengths and limitations of this study.

Good luck

Author Response

1) I think the abstract is too long. Please mention most important issues on the abstract. For example, answer categories of the questionnaires are not really important in the abstract.
Thank you very much. We have shortened the abstract.

2) Introduction needs to be extended, mostly using reporting the previous studies regarding the levels of PA in children with CHD as well as the effects of some important psychological factors on participation of children with CHD in PA.
Thank you very much. We added more information from the line 56 till 66:

In general, only 12% of children and adolescents achieved the WHO recommended physical activity of at least 60 min/day (15). In comparison, only 8.8% of children and adolescents with AHF achieved these recommendation (8). The lack of physical activity has several causes. Bjarnason-Wehrens (24) posited the "vicious cycle" as an explanatory approach. The lack of physical activity is created by overprotection and anxiety in the family of the affected person. This is reinforced by motor deficits. Motor deficits, in turn, increase anxiety and overprotection in the family. A "vicious cycle" develops. In addition, psycho-social situations become more difficult and the family's radius of action is limited. To address the problem of overprotection, there are recommendations for as-sessing physical fitness of children with CHD and detailed recommendations for physical activity even for competitive athletes cith Cardiovascular abnormalities (22, 23, 25, 26)

And also line 79-89:
The body of studies on physical self-concept in children with congenital heart defect is very limited (28, 29, 30). Only the studies by Chen et al. (29) used a questionnaire specific to physical self-concept, the Physical Self-Description Questionnaire (PSDQ). In the other two studies (28,30), physical self-concept was asked as a subscale in a questionnaire used for general self-concept.
Chen et al. (29,30) only made comparisons among children with AHF. Thereby, both studies found a significantly better physical selfconcept among boys compared to girls.
Chen et al. (28) analyzed the difference between school children with AHF and healthy school children. They found significantly lower physical self-concept scores in the children with AHF compared to the healthy children. However, due to its small sample size, the study by Chen et al. (29) is only of limited significance for the overall population.

3) I think it is better to place the tables and figures in right place (i.e., where the data are talked about).
We fully agree and have moved the figures and graphs to match the results in the body text.

4) Please explain about the strengths and limitations of this study.
Thank you very much, we have added a section discussing strenghth and limitation (please see line 298-326)
Strength and limitation
For this study a questionnaire was chosen to collect the data. Questionnaires are suitable for quantitative interviewing because they are associated with a low effort and costs. The disadvantage of questionnaires is that they are rather general and in contrast to the interview less detailed and can not respond to the individual. In return, the in-fluence of unintentionally and unconsciously dealing with each person differently is eliminated, resulting in deviations in measurement accuracy. Furthermore, the psy-chological effect of social desirability can lead to falsification of the answers. The CHD patients were contacted via online questionnaire and therefore had no opportunity to ask 
comprehension questions directly. To counteract this, participants had the option of calling the study leader or documenting comprehension questions at the end. For more accurate physical activity measurement data, directly measured physical activity using an accelerometer, for example, would be preferable but could (21). However, this would lead to a significantly higher financial outlay and a smaller sample size would be the consequence. Questionnaires are therefore a suitable means of survey for large samples for this research question.
One aspect that was not surveyed in this work is BMI. Due to the high number of overweight and obesity in CHD patients (Pinto et al., 2007; Tamayo et al., 2015) and the known influence on body perception and physical self-concept in healthy children and adolescents (Fernández-Bustos et al., 2019), it is an important parameter in terms of physical self-concept, but could not be considered in this work.
This work is a cross-sectional analysis. Thus, causal relationships cannot be de-rived. However, a cross-sectional analysis is sufficient for the presentation of descrip-tive information.
It cannot be ruled out that the reference group also includes sick children. How-ever, since this concerns only a minority, there should be no bias from a statistical point of view.
Due to the large sample size and the cooperation with the NRAHF and the Mo-torik Modul study representative data for Germany are presented.

Reviewer 2 Report

Congratulations to the authors on a well-designed study which considers an important topic in current practice – why children with CHD do less physical activity. I really enjoyed reading it and the results are very relevant for our patients.

Introduction

Some important references for exercise in children with CHD which I think would be relevant to include:

Takken et al Eur J Prev Cardiol 2012

Budts et al Eu Heart J 2020 (although for 16 yrs and older)

Van Hare et al Circ 2015

Materials and Methods

Well-designed study which makes use of excellent resources available from previous studies.

Results

Large patient cohort an obvious strength of the study.

Results are well presented and important similarities  between the cohorts highlighted (e.g. parental education levels).

Note than line 95 (results section) the reference is incorrect in that reference 16 is the Warnes reference but it is noted as reference 15 in this line.

Table 7: Frequency of a positive self-concept in the CHD group (in percent)

 I am not sure if the version I am seeing is correct. The X axis appears insufficiently labelled in that I cannot see what each of the percentage points correlates to – should it be a different skill? I only see flexibility noted on the x axis.

Table 9: The labelling on this table is also unclear in the version of the paper I am reviewing. The Y axis is not labelled. The X axis labelling is missing some letters (e.g. ‘n’ in Coordination; ‘e’ in Endurance’) and the colour code for the 3 different lines is missing.

Discussion and conclusion

Important points all highlighted.

Very important paper. The unexpected results regarding children with simple CHD performing significantly less exercise are of particular interest and like all good research, it is a paper which leads to more questions.

Author Response

Congratulations to the authors on a well-designed study which considers an important topic in current practice – why children with CHD do less physical activity. I really enjoyed reading it and the results are very relevant for our patients.

Thank you very much!

Introduction
Some important references for exercise in children with CHD which I think would be relevant to include:
Takken et al Eur J Prev Cardiol 2012
Budts et al Eu Heart J 2020 (although for 16 yrs and older)
Van Hare et al Circ 2015

Thank you very much for this important references, we have included these which match perfect to the content (please see reference number 22, 23 and 26)

Materials and Methods
Well-designed study which makes use of excellent resources available from previous studies.
Thank you!

Results
Large patient cohort an obvious strength of the study.
Results are well presented and important similarities between the cohorts highlighted (e.g. parental education levels).
Note than line 95 (results section) the reference is incorrect in that reference 16 is the Warnes reference but it is noted as reference 15 in this line.
Thank you very much for this note, this was a mistake and we correct it.

Table 7: Frequency of a positive self-concept in the CHD group (in percent)
I am not sure if the version I am seeing is correct. The X axis appears insufficiently labelled in that I cannot see what each of the percentage points correlates to – should it be a different skill? I only see flexibility noted on the x axis.We apologize for this, the graphics do not seem to be mapped correctly. For this reason we have reinserted them as a graphics file so that nothing can shift.

Table 9: The labelling on this table is also unclear in the version of the paper I am reviewing. The Y axis is not labelled. The X axis labelling is missing some letters (e.g. ‘n’ in Coordination; ‘e’ in Endurance’) and the colour code for the 3 different lines is missing.

We apologize for this, the graphics do not seem to be mapped correctly. For this reason we have reinserted them as a graphics file so that nothing can shift.
Discussion and conclusion.

Important points all highlighted.
Very important paper. The unexpected results regarding children with simple CHD performing significantly less exercise are of particular interest and like all good research, it is a paper which leads to more questions.
Thank you very much!